Data trace as the scientific foundation for trusted metrological data: a review for future metrology direction

Cao Zhanshuo 1
Gao Boyong 1
Liu Zilong 2 3 liuzl@nim.ac.cn
Xiong Xingchuang 2 3
Wang Bin 2 3
Pei Chenbo 2 3
1 China Jiliang University , Hangzhou, Zhejiang , China
2 Center for Metrology Scientific Data, National Institute of Metrology , Beijing , China
3 Key Laboratory of Metrology Digitalization and Digital Metrology, State Administration for Market Regulation , Beijing , China
Maguitman Ana
Electronic publication date: 2025 Aug 14
Publication date: 2025
Volume: 11
Electronic Location ID: e3106
Received 2024 Dec 23; Accepted 2025 Jul 15
Copyright: © 2025 Cao et al.
Copyright year: 2025
Copyright holder: Cao et al.
License: This is an open access article distributed under the terms of the Creative Commons Attribution License, which permits unrestricted use, distribution, reproduction and adaptation in any medium and for any purpose provided that it is properly attributed. For attribution, the original author(s), title, publication source (PeerJ Computer Science) and either DOI or URL of the article must be cited.
License URL: https://creativecommons.org/licenses/by/4.0/

Keywords: Data trace, Digital metrology, Tamper detection, Data forensics, File system, Data storage

Funding: National Key R&D Program of China 2023YFF0616904 National Science and Technology Infrastructure Platform Project APT2401-8 Key Project of Basic Scientific Research Business Expenses of the National Institute of Metrology, China AKYZD2404-3 This research was funded by the National Key R&D Program of China (Grant No. 2023YFF0616904), the National Science and Technology Infrastructure Platform Project (Grant No. APT2401-8), and the Key Project of Basic Scientific Research Business Expenses of the National Institute of Metrology, China (Grant No. AKYZD2404-3). The funders had no role in study design, data collection and analysis, decision to publish, or preparation of the manuscript.

==============================
In the context of the digital transformation of metrology, ensuring the trustworthiness and integrity of measurement data during its generation, transmission, and storage—i.e., trustworthy detection of measurement data—has become a critical challenge. Data traces are residual marks left during the data processing, which help identify malicious activities targeting measurement data. These traces are especially important when the trust and integrity of potential data evidence are under threat. To this end, this article systematically reviews relevant core techniques and analyzes various detection methods across the different stages of the data lifecycle, evaluating their applicability and limitations in identifying data tampering, unauthorized access, and anomalous operations. The findings suggest that trace detection technologies can enhance the traceability and transparency of metrological data, thereby providing technical support for building a trustworthy digital metrology system. This review lays the theoretical foundation for future research on developing automated anomaly detection models, improving forensic techniques for data tampering in measurement devices, and constructing multi-modal, full-lifecycle traceability frameworks for measurement data. Subsequent studies should focus on aligning these technologies with metrological standards and verifying their deployment in real-world measurement instruments.

Introduction

Metrology, as a fundamental discipline that ensures the accuracy, reliability, and consistency of measurement results in scientific and industrial applications, serves as a vital support for the national strategic framework (Barbosa et al., 2022). With the continuous advancement of metrological science, digital transformation has become a key pathway to improving the efficiency and precision of measurement processes (Eichstädt, Keidel & Tesch, 2021; Toro & Lehmann, 2021; Rab et al., 2023). This transformation leverages modern technologies to enable the digital processing of measurement data, thereby enhancing the efficiency and accuracy of data acquisition, analysis, and management. In this context, scientific data in the field of metrology plays a crucial role in the development of foundational data standards, with its security primarily reflected in the trustworthiness of the data. Only reliable metrological scientific data can be used to establish standard reference datasets and effectively support the national measurement standards through secure and trustworthy data sharing (Xiong et al., 2021).

Against this background, the concept of trustworthy detection of metrological data has emerged. Although there is currently no unified definition of data trustworthiness in the field of metrology, by integrating research findings from metrology, information technology, and data science (Lazzari et al., 2017; Velychko & Gordiyenko, 2019; Santos, 2020), it can be broadly understood as the use of technical means to verify and ensure the integrity and trustworthiness of metrological data throughout its entire lifecycle. This process involves verifying data sources, recording data changes, and performing integrity checks during data transmission (Kuster, 2021). The core objective of trustworthy detection is to ensure that any abnormal intervention during data acquisition, transmission, or storage can be promptly identified, with the system providing a clear and traceable audit trail (Liu et al., 2024).

It is worth noting that in the field of judicial forensics, a range of digital trace detection technologies has been proposed to meet data traceability requirements, and these technologies have gradually become key analytical tools in digital forensics (Horsman & Errickson, 2019). The core technologies aim to reveal traces of data generation, modification, and access behaviors, thereby providing a basis for assessing data trustworthiness and determining accountability. This concept aligns closely with the objectives of trustworthy detection in metrology. However, despite the logical and practical similarities between the two, the application of digital trace detection technologies in the field of metrology remains limited (Bianchi, Giannetto & Careri, 2018; Samoilenko & Tsiporenko, 2023).

Therefore, this article presents a systematic review of data trace detection technologies, summarizing the current state of research in this field. Building on existing studies of tamper detection, this article aims to provide valuable references for both data science researchers and metrology experts, while also identifying key research priorities and future development directions for trustworthy detection of metrological data. The main contributions of this article include: Key challenges: Summarize the key challenges encountered in the practical application of metrology and the main obstacles faced in the trustworthy detection of metrological data.

Analysis of existing research: An overview of the current advancements in data trace technologies is provided, establishing a foundational understanding of data traces and their role in trustworthy data detection.

Detailed review and classification: An in-depth analysis is conducted on the formation and evolution of data traces at various stages of data storage. Relevant literature is categorized based on different stages of the storage process, presenting the implementation methods of tampering detection technologies at each stage to offer a comprehensive understanding of the application characteristics of data trace technologies.

Future directions: The potential applications of data trace technologies in the trustworthy detection of metrological data are explored, providing guidance for the development of more robust systems for data protection and tampering detection in the metrology domain.

“Key Challenges” presents the major challenges faced in trustworthy detection in metrology; “Related Conceptual Knowledge” introduces the conceptual knowledge related to data traces; “Methodology” reviews the various stages of the data storage process; “Analysis of Existing Research” analyzes the research and technological advancements in data traces at different stages; “Limitations and Future Directions” focuses on the limitations and application prospects of data trace technology in the metrology field; “Conclusion” provides a summary of the article.

Key challenges

Although digital measurement methods offer significant advantages in terms of speed and accuracy compared to traditional methods, these advantages are facing serious challenges due to the rapid increase in the volume and variety of metrological data and the emergence of deep forgery technologies (Hasan et al., 2022). It has become increasingly difficult to ensure that digital data remains unaltered and complete, which not only directly affects the accuracy of metrological data but may also lead to severe economic losses. For instance, common metering instruments such as fuel dispensers (Melo et al., 2021; Almeida, Oliveira & Melo, 2024) and electric meters (Jacob & Samuel, 2020) are often targeted for illegal tampering.

Currently, the primary technical approaches for ensuring the trustworthiness of metrological data include digital signatures (Neyezhmakov, Zub & Pivnenko, 2021, 2022; Mustapää et al., 2022), hash function verification (Gregório, Miller & Duncan, 2024), and blockchain technology (Melo et al., 2021; Takegawa & Furuichi, 2023). These methods aim to guarantee the integrity and authenticity of key elements such as original file information, signature parameters, and timestamp data. Among them, the application of these technologies to trusted timestamping has become a commonly adopted method in the field of metrology for achieving data reliability and traceability (Matsakis, Levine & Lombardi, 2018; Melnik, Petrova & Vikulin, 2023).

Metrological scientific data typically contain a timestamp field that represents the time of the measurement activity during which the data were generated. This field serves both as metadata and as a trusted element for data traceability. Its reliability can be ensured through a well-structured and continuous time traceability chain. Trusted timestamping technologies are generally implemented based on high-precision time synchronization mechanisms and cryptographic signature algorithms, emphasizing the measurability and traceability of time information (Kumar et al., 2022; Wendt, 2025). These technologies ensure that the recorded time is anchored to a clear physical time reference and carries legal validity. Current research is advancing the integration of high-precision time sources into timestamping systems to further enhance their accuracy and verifiability.

In addition, some studies are exploring the integration of timestamping mechanisms with blockchain technology (Milicevic et al., 2022), leveraging distributed ledger structures to record the generation and verification processes of timestamps. This approach further enhances anti-tampering capabilities and operational transparency, providing critical support for the development of a trustworthy and highly reliable metrological data assurance system.

Although the aforementioned technologies have shown initial success in tamper prevention and data traceability during the data acquisition stage, a systematic, end-to-end trustworthy assurance framework for metrological data has yet to be established. In particular, there is still a lack of effective technical mechanisms to support post-tampering behavior detection, tampering path analysis, and accountability tracing. This shortcoming has become a critical obstacle hindering the further development of a trusted metrology system.

Related conceptual knowledge

Digital forensics

Digital forensics (DF), originating from forensic computer science, aims to encompass all digital devices that can be investigated. The National Institute of Standards and Technology (NIST) defines DF as “the scientific application of data identification, collection, examination, and analysis, while ensuring the integrity of information and maintaining a strict chain of custody” (Lyle et al., 2022). Due to its multidisciplinary and interdisciplinary nature, DF can be widely extended to other fields. For example, devices used in metrology are also potential sources of digital evidence (Irons, 2010).

In modern digital forensics investigations (see Fig. 1), the core component of ensuring data security is the “Ensure and Detect” phase. The main function of this phase is to help determine whether unauthorized operations have occurred and to ensure data security through relevant technologies and tools. In this process, the key factor in identifying and tracking data anomalies is data trace.

Figure 1 Steps of digital forensics investigation.

Data trace

The concept of data traces originates from the interpretation of physical tool marks in forensic science. Similar to the marks left by physical tools on the surface of objects (Kaur, Sharma & Sharma, 2024), data also leaves distinct traces during storage and processing. As Kessler (2007) demonstrates, “most processes leave identifiable erasure traces,” indicating that when unauthorized users attempt to manipulate or tamper with data through computer system operations, these actions are likely to leave detectable traces.

Data traces typically encompass various types of operation-related information, such as operating system usage files, specific tool footprints, system-generated logs, and file system metadata. This information plays a critical role in recording and reflecting operational activities. Even when certain tools attempt to overwrite or conceal traces of operations, residual logs, caches, or hidden metadata within the system may still retain partial data traces, thereby providing key evidence for digital trust verification and provenance analysis. These traces not only reveal the time, location, and nature of operations but also potentially uncover details of tampered or deleted data, thereby supporting the detection of data trustworthiness and integrity breaches. Furthermore, they assist digital investigators in tracking the evolution of data, reconstructing timelines of data operations, and analyzing the causes of data loss, thus offering significant value.

Anti-forensics

Anti-forensics (AF) is a frequently mentioned term in the field of DF, referring to various techniques aimed at weakening, concealing, tampering with, or deleting digital evidence (González Arias et al., 2024). The primary objective of these techniques is to disrupt, obscure, or fabricate data traces, thereby hindering the process and accuracy of DF investigations. Although no formal and unified definition currently exists, most practitioners and scholars generally agree that any attempt to alter, interfere with, negate, or obstruct scientifically valid forensic investigations through technical means can be classified as AF techniques (Mohammad et al., 2024).

For example, timestamp forgery is a common AF technique that involves modifying a file’s creation, modification, or access times in order to confuse investigators and complicate or invalidate timeline analysis (Oh, Lee & Hwang, 2024). Common tools for timestamp forgery are shown in Table 1. In practice, the impact of AF techniques extends beyond the field of DF. By compromising data trustworthiness and traceability, these techniques pose a threat to any field that relies on high data credibility.

Table 1 File system timestamp tampering tools.

Forgery methods	Forgery tools	
API function (second-level precision)	Dat FileTouch, NewFileTime, SKTimeStamp, Bulk FileChanger, Change Timestamp, and eXpress TimeStamp Toucher	
API function (nanosecond-level precision)	Timestamp, nTimestamp	
Disk access	SetMace1	
Note:

1 Starting from version 1.0.0.6, SetMACE adopts direct disk access.

In practical applications, the forms and technical methods of AF can be diverse and are not limited to digital tools or methods specifically designed for this purpose, exhibiting increasingly complex characteristics. In fact, certain techniques and actions not originally intended for AF can inadvertently impact data integrity and trustworthiness (Palmbach & Breitinger, 2020; Yaacoub et al., 2021), especially in areas related to privacy protection and system optimization. For example, disk defragmentation or automatic cleanup tools are designed to improve system performance or protect user privacy, yet these technologies may delete or overwrite data valuable to digital investigations. Although these tools lack explicit AF intent, they can still hinder digital forensic efforts.

The key factor in distinguishing between legitimate actions and malicious AF activities often lies in the user’s subjective intent. Many privacy-enhancing technologies, such as encryption, steganography, and anonymous communication, play a significant role in protecting user privacy. However, when maliciously exploited, these technologies can severely hinder the verification of data trustworthiness, thereby further complicating the field of data security.

Data storage process

After years of development, DF has been widely applied in various fields. In the context of metrology, to effectively utilize DF, it is essential to focus on the aspects most pertinent to metrologists—the data transmission and storage processes in metrological devices. This focus serves as the basis for the classification in our current study.

Currently, to ensure data security, the vast majority of metrological data storage still relies on physical storage. The general process of data storage is illustrated in Fig. 2: users interact with applications through the operating system’s user interface, and the input data is then transmitted from the application to the operating system. Within the operating system, data is managed by the file system and ultimately passed to the storage system. The operating system, in conjunction with the host controller and storage controller, writes data to the physical storage.

Figure 2 The data storage process.

During this process, the data undergoes multiple transformations, transitioning from a high-level, user-understandable format to a binary format suitable for physical storage, as shown in Fig. 3. Clearly, due to the inherent instability of computer data, any operation can lead to changes in the metadata. Furthermore, each stage of the data processing flow exhibits distinct characteristics and operational methods, which results in different focal points for data trace analysis at different stages.

Figure 3 The transformation of data forms.

Methodology

To comprehensively organize and summarize the research efforts in the field of data traceability, this article outlines the following research plan: Literature review by stages: The types of data traces generated at different stages of the data storage process vary, as shown in Table 2. Therefore, to better understand the different data trace detection techniques at each stage, this study analyzes relevant research and technologies in stages based on the data storage process, providing a detailed discussion of the key issues and research advancements related to data tracking at each stage.

Comprehensive literature search: A comprehensive literature review was conducted on research related to data trace detection in the field of digital forensics, encompassing several authoritative databases, including SpringerLink, ScienceDirect, IEEE Xplore, Google Scholar, and Web of Science, to ensure a broad and diverse range of resources. In addition, we manually examined the reference lists of the selected studies and analyzed their citation records to identify potential relevant research.

Rigorous selection criteria: To ensure the quality and rigor of this review, the retrieved studies were subjected to detailed screening and quality evaluation. Studies unrelated to the topic of data traceability and those published in low-quality journals were excluded. Priority was given to research published in high-impact journals and top-tier conferences to guarantee the authority and academic value of the review.

Information extraction from selected studies: Key information, including authorship, publication year, research type, and major contributions, was extracted and organized into tables or visual formats to facilitate comparisons and analyses for scholars in the metrology field engaging in interdisciplinary studies.

Table 2 Data traces across different stages.

Stage	Data traces	
Application layer	Multimedia features, application logs, configuration files	
Operating system	OS logs, process traces, network activity, system calls, error reports	
File system	File allocation table, metadata (timestamps, permissions, owner, size), file paths	
Storage device	Disk fragments, memory caches, virtual memory, deleted files	

Based on the aforementioned methodology, this study conducted a literature search using keywords related to digital evidence, anti-forensics, and digital forensics, in combination with terms related to “data traces” (such as detection, traces, and tampering). A total of 1,937 articles published in the field of data forensics over the past five years were identified, among which 202 articles contained terms related to data traces. Subsequently, these 202 articles were screened, eliminating irrelevant and duplicate articles, and citation records were further analyzed to identify additional relevant studies. To ensure comprehensiveness, articles that were highly relevant but not captured through automated searches were manually included. In the end, 81 articles were selected as the analysis objects for this study, and the keyword frequency distribution of the selected articles is shown in Fig. 4.

Figure 4 Word cloud of keywords from related literature.

Figure 5 illustrates the year distribution of the selected articles, indicating that most of the publications were published within the last five years. Based on the content of the research, this study categorizes the literature into four categories: application layer, operating system, file system, and storage devices. Figure 6 presents the distribution of the number of articles in these four categories, showing that the output of articles across these categories is relatively balanced, with each category having a substantial number of articles published in the past five years.

Figure 5 Year distribution of literature on data traces.

Figure 6 Distribution of literature across research categories.

Analysis of existing research

Application layer trace detection

Prior knowledge

The application layer is an advanced stage of data processing, primarily responsible for high-level data management and operations. At this stage, the data has typically undergone initial encoding, decoding, and transformation, making it meaningful and usable. Multimedia data, such as images, audio, and video, are common data types used at this stage. For example, in the field of metrology, digital calibration certificates (Neyezhmakov, Zub & Pivnenko, 2021) and industrial measurement inspection images (Lins, Santos & Gaspar, 2023) are part of multimedia data.

Existing multimedia data tampering operations typically include deleting, hiding, or replacing specific regions, adding new elements, and adjusting the size or position of existing elements. Although these tampering operations may leave no visible or audible traces through anti-forensic techniques, the data’s underlying structure, encoding methods, and compression characteristics often still reflect traces of modification. Through specific detection techniques, these traces can generally be identified and restored, revealing whether the data has been tampered with or forged (Farid, 2016).

Related research

Currently, the majority of research focuses on multimedia data tampering detection, covering areas such as text, images, audio, and video. Additionally, some studies are dedicated to the detection of anti-forensic attacks. The literature related to application-layer data traces is summarized in Table 3.

Table 3 Trace monitoring related literature at the application layer.

Related research	Research content	Contribution	
Bourouis et al. (2020)	Multimedia	Reviewing the latest advancements in digital multimedia tampering detection while proposing an intuitive soft classification method to enhance the reliability of digital multimedia authentication.	
Wang et al. (2022)	Multimedia	Reviewed current media tampering detection methods, including image tampering detection and Deepfake detection, and discussed the challenges and future research trends.	
Zhang, Wang & Zhou (2018)	Video	Summarized the current research progress in video forgery detection technologies, covering various active and passive detection methods, and suggested future research directions.	
Shelke & Kasana (2021)	Video	Conducted a comprehensive survey on passive techniques for digital video forgery detection, analyzing existing methods, discussing anti-forensic strategies and deepfake video detection, and proposed a standardized benchmark dataset and a general framework for digital video forgery detection.	
Akhtar et al. (2022)	Video	Comprehensively reviewing existing passive video tampering detection techniques, analyzing the strengths and weaknesses of current research, and discussing the limitations of video forensics algorithms along with future research directions.	
Li & Huo (2024)	Video	A video frame deletion detection method based on continuity-attenuation features is proposed, capturing spatiotemporal characteristics of forged videos through a deep learning network architecture. Experimental results show a detection rate of 93.85%.	
Shaikh & Kannaiah (2025)	Video	A hybrid approach combining traditional block-based analysis and convolutional neural networks (CNNs) for video forgery detection, effectively addressing advanced forgery techniques. The method achieved an accuracy of 79.31% and an F1-score of 65.87, outperforming existing methods.	
da Costa et al. (2020)	Image	Reviewed the current state of research in image tampering and anomaly detection, identified future opportunities and challenges, and provided an innovative contribution through in-depth analysis and comprehensive evaluation of over 100 related studies.	
Padilha et al. (2022)	Image	Proposed a content-aware method for detecting timestamp manipulation by verifying the consistency between image content, capture time, and geographic location. The approach significantly improved classification accuracy on a large benchmark dataset and can estimate capture time when the timestamp is missing.	
Bharathiraja et al. (2024)	Image	A comprehensive survey on image forgery techniques and detection methods, discussing source camera identification, type-dependent and independent methods, and deep learning approaches. Aims to enhance understanding of image forgery detection and demonstrate cross-domain applications.	
Kaur, Jindal & Singh (2024)	Image	A review of passive image forgery detection techniques, discussing common forgeries like splicing, copy-move, and retouching, along with frameworks, taxonomy, and methodologies. Aims to assist researchers in overcoming challenges and offers future directions.	
Capasso, Cattaneo & De Marsico (2024)	Image	A survey on image integrity detection methods, discussing common forgery attacks and countering techniques, evaluating the limitations of existing methods in detecting counterfeit images. Highlights the potential of combining approaches, particularly convolutional neural networks, for forgery detection.	
Johnson & Davies (2020)	Text	Used digital forensic techniques to analyze a contract cheating case, revealing inconsistent editing patterns through revision identifiers in Microsoft Word documents. Developed a tool using visualization techniques to identify fraudulent editing traces, aiding in the detection of contract cheating.	
Johnson, Davies & Reddy (2022)	Text	Explored the application of digital forensics techniques in higher education for detecting academic misconduct, proposing the repurposing of tools used in criminal cases for file tampering and metadata extraction to detect plagiarism and contract cheating. Developed a prototype software tool for automated plagiarism/contract cheating detection.	
Joun, Lee & Park (2023a)	Text	Proposed a novel framework for precise tracing of deleted or overwritten data through data remnants analysis, overcoming the limitations of traditional digital forensics methods. Case study on Microsoft 365 demonstrated superior efficacy and accuracy of the approach.	
Spennemann, Spennemann & Singh (2024)	Text	Examined how RSID numbers in Microsoft Word can be used for detecting academic misconduct. Explored how common editing actions like copy, paste, and text insertion leave traces in the rsid encoding, and provided prompts to guide investigators in interviewing alleged perpetrators.	
Leonzio et al. (2023)	Audio	Proposed a method for detecting and localizing audio splicing based on CNN, using model-specific features to identify if an audio track has been spliced. Manipulations are detected through clustering, and splicing points are located using a distance-measuring technique.	
Son & Park (2024)	Audio	Proposed a deep learning-based approach for detecting forged audio files, specifically targeting modifications made using the “Mixed Paste” command. The method combines a convolutional neural network and a long short-term memory (LSTM), utilizing Korean consonant features from spectrograms, achieving a high accuracy rate of 97.5%.	
Ustubioglu et al. (2025)	Audio	Proposed a novel audio copy-move forgery detection method, identifying forged segments by extracting high-resolution spectrograms and matching keypoints, combined with graph-based representations and CNN classification, achieving superior detection accuracy. The method performs well across various attack scenarios, demonstrating strong generalization potential.	
Rekhis & Boudriga (2011)	AF	Proposed a theoretical approach to digital forensic investigations that considers anti-forensic attacks, addressing the challenge of attackers obstructing event analysis and attack scenario reconstruction through such attacks.	
Rekhis & Boudriga (2012)	AF	Proposed a novel hierarchical visibility theory for digital investigations to counter anti-forensic attacks, enabling the handling of security events in complex systems and improving the provability of attacks.	

In summary, current application-level data trace detection methods can be classified into two main categories: active detection and passive detection, as shown in Fig. 7. Active detection methods rely on auxiliary information, such as digital watermarks, digital signatures, and revision save identifiers (RSID), to ensure the authenticity and integrity of the data. These methods embed additional markers or information into the data, enabling timely detection in the event of tampering. However, a limitation of active detection methods is that they require pre-embedded anti-tampering mechanisms, and in some cases, an attacker may be able to remove or modify these identifying markers. In recent years, blockchain technology has received increasing attention in the field of active detection. Its decentralized and immutable features effectively enhance data security (Sakshi, Malik & Sharma, 2024; Ramezanzadehmoghadam, 2024); however, since this study focuses on detecting traces of tampered data, blockchain technology is not the primary focus of this article.

Figure 7 Classification of multimedia tampering detection.

In contrast, passive detection methods do not rely on any pre-embedded information, but rather analyze the inherent characteristics of the data itself. These methods detect tampering by exploiting inevitable inconsistencies during the tampering process, such as pixel-level anomalies, geometric distortions, physical lighting inconsistencies, and compression artifacts. Since passive detection methods do not require special processing during data generation, they offer greater flexibility and are less susceptible to tampering by attackers. However, the detection accuracy of these methods typically depends on the quality of the data. When facing issues such as blurred data edges, compression, or noise, the performance of the technology may degrade, leading to a higher false positive rate.

Researchers are actively exploring the use of emerging technologies, such as deep learning (Padilha et al., 2022; Leonzio et al., 2023; Shaikh & Kannaiah, 2025), to enhance detection capabilities. Although deep learning methods have made significant progress in certain applications, the existing methods typically use different evaluation metrics and datasets, making comparisons between them difficult. Additionally, while improving detection accuracy helps identify more tampered behavior, it often increases computation time and resource consumption, which presents a significant challenge in current research. Therefore, balancing detection accuracy, time efficiency, and resource consumption remains a critical issue that future application-level research must address.

Operating system trace detection

Prior knowledge

Operating systems generate and manage various metadata during data transmission, typically including file operation records, user activities, and system states. However, different operating systems exhibit significant differences in data processing, recording methods, and log management strategies. The major operating systems, as shown in Fig. 8, can generally be divided into two branches: one based on Microsoft Windows and the other derived from Unix. The Windows NT series features a complex event logging system that comprehensively records file operations, user logins, and other activities. In contrast, Unix/Linux systems typically rely on syslog to log system behaviors, resulting in a narrower scope of logs compared to Windows systems. macOS combines a Unix-style logging system with its own activity monitoring tools, enabling the capture of a broad range of system and user activity information. Therefore, understanding the data processing methods and log recording mechanisms of different operating systems is crucial for effective digital trace detection.

Figure 8 Classification of operating systems.

Related research

In recent years, researchers have conducted extensive studies on trace detection at the operating system level, covering most mainstream operating systems. Table 4 summarizes the literature related to operating system data traces.

Table 4 Trace monitoring related literature at the operating system layer.

Related research	Research content	Contribution	
Lees (2013)	Windows	Explored methods for detecting the deletion of digital evidence using Update Sequence Number (USN) log files, identifying recognizable patterns left by specific software usage and ‘private browsing’ modes.	
Singh & Gupta (2019)	Windows	Studied file timestamp patterns generated when using different operating system tools to manipulate files in the Windows Subsystem for Linux environment on Windows 10, aiming to detect timestamp forgery.	
Studiawan, Sohel & Payne (2019)	Windows	Provided a comprehensive overview of forensic analysis techniques, tools, and datasets for operating system logs, and proposed potential directions for future research.	
Dija et al. (2020)	Windows	Proposed a detailed process for reconstructing or retrieving key information related to suspicious programs from multiple sources in Windows systems, even if the programs have been deleted or overwritten.	
Joo, Lee & Jeong (2023)	Windows	Explored the impact of file-wiping tools on different Windows artifacts, analyzed traces left by 10 file-wiping tools in 13 Windows artifacts, and created a reference database to assist forensic investigators in identifying the use of file-wiping tools.	
Vanini et al. (2024)	Windows	Introduced the concept of time anchors to validate and correct system clock skew, addressing timestamp inaccuracies caused by system clock deviations. Demonstrated the use and benefits of time anchors in digital forensic event reconstruction through specific case examples.	
Todd & Peterson (2024)	Windows	Introduced digital trace inspector (DTI), a learning classifier system (LCS)-based tool for temporal metadata analysis. The tool groups temporal digital traces of targeted user activity using a binary Michigan-style LCS and constructs rules through an expert knowledge framework. Experimental results show perfect recall across 10 user behavior scenarios, with an average F1-score of 0.98 and minimal training data required.	
Garland et al. (2024)	Windows Linux	Investigated digital forensic artifacts related to 3D printing slicing software, analyzing data stored in Windows and Linux operating systems. By collecting RAM, configuration data, generated files, residual data, and network data, vital evidence was identified for 3D printing forensics.	
Fiadufe et al. (2025)	Windows	Investigated malware forensics in Windows 7 OS, using memory and disk forensics tools to analyze malicious software through static and dynamic analysis. Proposed a comprehensive malware forensics framework to enhance malware detection and analysis.	
Thierry & Müller (2022)	Unix	Proposed a comprehensive approach to deeply analyze MACB timestamps in Unix-like systems, revealed the influence of different layers of the software stack, and introduced a framework for automatic analysis of operating system kernels, user-space libraries, and applications, while identifying and documenting various unexpected and non-POSIX-compliant behaviors.	
Govindaraj et al. (2015)	IOS	Introduced the iSecureRing solution, which provides security for mobile applications on jailbroken iPhones and preserves event timestamps to support forensic analysis.	
Joun, Lee & Park (2023b)	macOS	Proposed a method to track deleted files by analyzing various sources of file-related metadata in macOS systems, addressing the issue of spoliation of evidence, particularly in cases of information leakage, digital sexual crimes, accounting fraud, and copyright infringement.	
Pieterse, Olivier & Van Heerden (2015)	Android	Proposed a new method called the Authenticity Framework for Android Timestamps (AFAT) to verify the authenticity of SQLite database timestamps in Android smartphones.	
Pieterse, Olivier & Van Heerden (2016)	Android	Proposed a reference architecture for detecting tampering evidence in Android applications, focusing on identifying evidence manipulation through the analysis of timestamps in SQLite databases.	
Pieterse, Olivier & Van Heerden (2017)	Android IOS	Proposed a novel reference architecture for smartphone applications and derived seven theories of normal behavior from this architecture to evaluate the authenticity of smartphone evidence.	
Pieterse, Olivier & Van Heerden (2019)	Android IOS	Investigated the impact of manipulating smartphone data on Android and iOS platforms and proposed an evaluation framework for detecting tampered data, successfully demonstrating data manipulation behaviors and providing preliminary evidence that the framework aids in detecting tampered smartphone data.	
Han & Lee (2022)	Android	Studied the handling of timestamps for multimedia files on mobile devices, examining how operating systems and filesystems manage various types of timestamps and the potential impact of timestamp reversal on digital forensics.	
Park et al. (2022)	Android	Proposed a forgery analysis method for Samsung smartphone call recordings, identifying tampering by analyzing audio bandwidth, latency, file structure, media-log, and call history to verify the authenticity of the recordings.	
Rathore et al. (2023)	Android	Investigated adversarial attacks and defenses in Android malware detection, introducing two reinforcement learning-based evasion attacks, analyzing the vulnerabilities of 36 detection models, and proposing the MalVPatch defense strategy to enhance detection accuracy and adversarial robustness.	
Valeti & Rathore (2024)	Android	Introduced a malware detection model based on Android permissions and intents, presented a novel adversarial attack method GBKPA, analyzed the transferability of adversarial samples, and proposed the AuxShield defense mechanism, significantly enhancing the adversarial robustness of detection models.	

In summary, trace detection at the operating system level encompasses multiple mainstream operating systems, with a primary focus on tampering detection in system logs and file metadata, particularly addressing core issues such as data deletion, timestamp forgery, and malware forensics. For instance, the temporal metadata analysis method proposed by Todd & Peterson (2024) demonstrated exceptionally high recall and accurate recognition of user behavior scenarios, particularly with significant breakthroughs in reducing the need for training data, which is crucial for enhancing forensic efficiency and accuracy. Additionally, the timestamp validation framework introduced by Pieterse, Olivier & Van Heerden (2015, 2016, 2017, 2019) for Android and iOS platforms greatly improved the detection of data tampering in mobile devices, showing strong adaptability and accuracy when dealing with timestamp forgery in databases.

However, despite these advancements, there remains a notable bias in the research on operating system-level data trace detection. Most of the research focuses on Windows and Android systems, while studies on Unix-like systems and iOS (particularly due to the constraints of closed ecosystems) are relatively scarce. Although some scholars (Garland et al., 2024) have made positive strides in cross-platform forensics, the challenges posed by the forensic limitations of closed ecosystems and the fragmentation of open systems still result in insufficient generalization and cross-platform compatibility of current technologies. Therefore, future research should place greater emphasis on improving the general applicability of existing technologies, particularly in both open and closed ecosystem environments, to address compatibility issues across different operating systems.

File system trace detection

Prior knowledge

The file system records various metadata associated with files, documenting the entire history of file operations, from creation and modification to deletion, and even enabling the traceability of a file’s origin and content changes. As such, it is a crucial source of data traces. Currently, there is a wide variety of file systems used across different fields, and for the same operating system, multiple file systems are typically available for selection, as detailed in Table 5. Different types of file systems generate varying amounts and types of metadata and utilize their own unique mechanisms to record and manage this metadata. In short, the complexity of the file system’s design directly affects the richness and accuracy of its metadata.

Table 5 Common operating systems and supported file systems.

Operating system	File system	
Windows	FAT12, FAT16, FAT32, exFAT, NTFS, ReFS	
Mac	APFS, HFS, HFS+, FAT32, ExFAT, UDF	
Linux	Ext, Ext2, Ext3, Ext4, Btrfs, XFS, JFS, GFS, VFAT, HPFS, Swap	
FreeBSD	UFS, ZFS, FAT32, ext2, ext3, ext4	
Android	FAT32, Ext3, Ext4, exFAT, F2FS	
iOS	APFS, HFS+, FAT32	
Optical media	ISO 9660, UDF (Universal Disk Format)	
Embedded systems	FAT12, FAT16, FAT32, ext2, ext3, JFFS2	

For example, the NTFS file system (Fig. 9), compared to the FAT file system (Fig. 10), records more detailed operational information and logs, thus generating richer metadata. The potential data traces produced by these file systems provide critical support for trace detection technologies, enabling researchers to more easily track the full lifecycle of data and effectively identify traces of file tampering and their origins (Buchholz & Spafford, 2004).

Figure 9 NTFS file system architecture.

Figure 10 FAT file system architecture.

Related research

Table 6 summarizes the key research findings related to file system trace detection.

Table 6 Trace monitoring related literature at the file system layer.

Related research	Research content	Contribution	
Chow et al. (2007)	NTFS	Focused on temporal analysis of the NTFS file system and proposed intuitive rules for digital file behavior characteristics to aid event reconstruction in computer forensics.	
Neuner et al. (2016)	NTFS	Proposed and studied the feasibility of using file timestamps as a steganographic channel, leveraging the information gap between storage and use of timestamps with high-precision timers in modern operating systems, designing a layered steganographic system with secrecy, robustness, and wide applicability.	
Neuner et al. (2017)	NTFS	Evaluated the steganographic potential of redundant capacity in file system timestamps for information hiding and data exfiltration, and proposed technical methods to assist digital forensic investigations in identifying and detecting tampered file system timestamps.	
Nordvik, Toolan & Axelsson (2019)	NTFS	Proposed a new method leveraging the $ObjId Index in the NTFS file system to record and correlate user activities, providing a novel tool for storage device forensics.	
Bahjat & Jones (2019)	NTFS	Proposed a framework for determining the time window of deleted file fragments by analyzing the timestamps of allocated neighbors, innovatively leveraging the known temporal states of adjacent clusters to infer the creation and deletion times of file fragments.	
Mohamed & Khalid (2019)	NTFS	Proposed a machine learning-based method for the automatic detection of NTFS file system timestamp tampering, improving detection efficiency and accuracy.	
Palmbach & Breitinger (2020)	NTFS	Proposed the use of four existing Windows system log files to detect timestamp tampering in the NTFS file system and analyzed the reliability of this evidence.	
Oh, Park & Kim (2020)	NTFS	Proposed an anti-anti-forensics method based on NTFS transaction characteristics and machine learning algorithms, capable of detecting data wiping traces, effectively identifying wiped files and the tools used, providing evidence for digital forensic investigations.	
Kao (2020)	NTFS	Explored timestamp transfer and contact artifacts in data hiding operations, proposing a forensic exchange analysis method based on temporal attributes, with practical applications in crime scene reconstruction. Experimental results demonstrate the method’s effectiveness in uncovering hidden relationships and tracing forensic evidence.	
Galhuber & Luh (2021)	NTFS	Investigated file timestamp patterns generated by user operations in the NTFS file system to address timeline forgery techniques commonly used in anti-forensics, and evaluated the effectiveness of seven third-party timestamp forgery tools along with the forensic evidence they produce.	
Horsman (2021)	NTFS FAT32	Studied eight free file wiping tools and identified the “Digital Tool Marks” (DTMs) left behind after using these tools, exploring whether these traces can be linked to specific wiping tools to enhance digital forensic analysis.	
Bouma et al. (2023)	NTFS	Proposed a method for reconstructing file history based on NTFS timestamps, innovatively inferring all possible file operation sequences and generating a tree-structured timeline of file history for digital forensics.	
Song & Lee (2023)	NTFS	Designed a timestamp manipulation detection method based on storage performance in NTFS, which verifies file write time and compares it with storage performance to detect timestamp tampering. This method overcomes the limitations of traditional detection techniques, preventing precise timestamp manipulation.	
Oh, Lee & Hwang (2024)	NTFS	Compared and analyzed existing NTFS timestamp manipulation detection methods, proposing a new detection algorithm based on NTFS journals to overcome the limitations of previous methods. Experimental results show significant improvements in detecting timestamp manipulations and identifying normal events.	
Lee & Song (2024)	NTFS	Proposed a method for detecting abnormal Windows file paths based on natural language processing, converting file paths into vectors and classifying them using machine learning. Experimental results demonstrated an accuracy of up to 94% in detecting abnormal file paths.	
Nordvik et al. (2021)	Compre- hensive	Proposed a formal reliability validation procedure for file system reverse engineering, focusing on documenting the forensic process and tools to ensure reliability and reproducibility. Highlighted the inadequacy of current validation methods in meeting legal and scientific requirements.	
Heeger, Yannikos & Steinebach (2021)	exFAT	Proposed two methods for hiding data within the exFAT file system, with exHide offering higher robustness and a reasonable embedding rate. The study evaluates both methods and discusses their strengths and weaknesses.	
Nordvik & Axelsson (2022)	exFAT	Investigated the handling of timestamps in the exFAT file system, revealing significant inconsistencies between Windows, MacOS, and Linux implementations, as well as discrepancies in forensic tools’ interpretation. Concluded that timestamp interpretation in criminal investigations should not assume flawless adherence to file system specifications or accurate interpretation by forensic tools.	
Lee et al. (2023)	ReFS	Proposed a method for detecting data wiping tools based on deletion patterns in the ReFS $Logfile, identifying specific deletion behaviors through opcode variations. Demonstrated the application and effectiveness of this method in digital forensics.	
Toolan & Humphries (2025)	XFS	Examined five methods of data hiding in the XFS file system, evaluating them based on capacity, detection difficulty, stability, and other metrics. Highlighted XFS as a viable option for data hiding.	

In the field of file system-level trace detection, timestamp analysis occupies a central position. Current research has gradually evolved from traditional temporal rule modeling to multi-dimensional detection techniques. However, experiments by Galhuber & Luh (2021) on seven types of third-party tampering tools indicate that existing detection methods still face the challenge of increasingly sophisticated falsification techniques. For example, attackers may exploit redundant fields in the lower layers of the file system to bypass traditional validation mechanisms, thereby evading detection.

Moreover, differences in file systems pose significant challenges to forensic reliability. Studies have shown that the characteristics of different file systems directly affect the effectiveness of detection. For example, the ambiguity in timestamp interpretation of exFAT across Windows, Linux, and macOS may lead to conflicting forensic conclusions across platforms (Nordvik & Axelsson, 2022).

Overall, current research faces three major bottlenecks: first, an overemphasis on NTFS, which results in the delayed development of detection methodologies for other file systems; second, detection algorithms are predominantly based on laboratory environments, lacking robustness validation in complex real-world scenarios; and third, there is a lack of evaluation of the trustworthiness of forensic tools themselves.

Storage device trace detection

Prior knowledge

Storage devices can be classified into two main categories: persistent storage and volatile storage (see Fig. 11). Persistent storage includes hard drives (HDD), solid-state drives (SSD), and flash memory, all of which are characterized by their ability to retain data for long periods even after power loss, through physical or electronic media. For example, an HDD relies on magnetic platters to store data, and data can sometimes be recovered even after the device has been formatted. In contrast, an SSD uses different storage and deletion mechanisms, and its data storage and erasure processes are unique. In some cases, an SSD may even support permanent data deletion, a feature that makes it easier for criminals to tamper with or delete data traces (Kumar, 2021). Additionally, there is no straightforward method to effectively prevent the recovery of data traces after an SSD has been erased.

Figure 11 Classification of storage devices: persistent vs. volatile storage.

In contrast to persistent storage, volatile storage refers to storage devices that lose data after power is removed, such as common DDR4 and DDR5 memory. Since volatile memory can only retain data while power is supplied, special care must be taken when collecting and analyzing data from volatile storage to avoid interfering with the contents of the memory during its operation. Trace analysis of volatile storage typically relies on memory dump techniques, which quickly extract data from memory to recover information such as program states and user activity logs (Nyholm et al., 2022). However, the dynamic nature of memory data and the loss of data upon power down make volatile storage trace analysis more challenging compared to persistent storage.

Related research

In recent years, research on storage device traces has mainly focused on memory acquisition techniques, and therefore, the studies summarized in Table 7 are primarily concentrated in the field of memory forensics.

Table 7 Trace monitoring related literature at the storage device layer.

Related research	Research content	Contribution	
Case & Richard (2016)	Memory	Focused on using memory forensic techniques to detect user-space malware written in Objective-C and proposed innovative methods for analyzing user-space malware.	
Pagani, Fedorov & Balzarotti (2019)	Memory	Introduced the temporal dimension in memory forensics, exploring the impact of acquisition time on analysis results, validating the negative effect of temporal inconsistencies through experiments, and modifying existing tools to explicitly account for the temporal dimension and minimize its effects.	
Schneider, Wolf & Freiling (2020)	Memory	Studied the difficulty of tampering with main memory images during digital forensics and the methods for detecting such tampering, revealing that while tampering with main memory images is challenging, it carries a significant probability of misleading analysts.	
Parida & Das (2021)	Memory	Proposed a mechanism called PageDumper to collect page table manipulation information at runtime, capturing attack footprints related to memory protection manipulation and supplementing memory snapshot data for more practical and in-depth memory analysis.	
Seo, Kim & Lee (2021)	Memory	Proposed a method to detect headerless executable files by scanning the Section table in memory dumps, addressing the issue of malware evading memory forensics by removing executable file headers, and implemented the method as a plugin in Volatility 3 framework.	
Manna et al. (2022)	Memory	Developed a suite of memory analysis plugins targeting.NET and.NET Core applications, automatically identifying key runtime areas and suspicious malicious components, enhancing the detection of modern malware, especially memory-only attacks.	
Arfeen et al. (2022)	Memory	Developed a framework for regular volatile memory acquisition to analyze individual process behavior, extracting key features and utilizing machine learning to efficiently classify malicious and benign processes, improving ransomware detection accuracy.	
Bellizzi et al. (2022)	Memory	Proposed Just-in-time Memory Forensics (JIT-MF) to address the challenge of capturing short-lived evidence on Android devices, developed the MobFor tool to capture evidence of messaging hijack attacks, outperforming traditional tools in real-world attack scenarios.	
Zhang et al. (2023)	Memory	Proposed a malware detection method based on convolutional neural networks and memory forensics, analyzing PE file fragments in memory, achieving high detection accuracy of 97.48%, particularly for fileless malware.	
Khalid et al. (2023)	Memory	Proposed a machine learning-based detection method for fileless malware, extracting features from memory dumps using Volatility tool and analyzing them with machine learning algorithms. Experimental results show Random Forest achieved 93.33% accuracy, outperforming other classifiers.	
Rzepka et al. (2024)	Memory	Evaluated causal inconsistencies in Windows 10 memory dumps using two methods, finding that a high level of inconsistencies is common, and proposed the possibility of controlling inconsistency levels by managing workload and the execution timing of memory acquisition tools.	
Parida, Nath & Das (2024)	Memory	Proposed a mechanism named SAM to support wipe-aware legal analysis in non-atomically acquired volatile memory snapshots, enhancing the reliability and quality of memory analysis.	
Kirmani & Banday (2024)	Memory	Conducted an in-depth analysis of the importance of reliability in physical memory forensics and proposed a comprehensive set of strategies and practical methods to enhance the reliability of memory forensics.	
Penrose, Buchanan & Macfarlane (2015)	HDD	Proposed a fast contraband detection method using disk sampling and Bloom filters, achieving 99.9% accuracy in scanning large-capacity disks in minutes, addressing the challenge of forensic tools lagging behind increasing storage capacities.	
Meffert, Baggili & Breitinger (2016)	HDD	Proposed an attack on digital forensic tools by modifying the firmware of a popular hardware write blocker (TD3), corrupting the integrity of the destination disk without the user’s knowledge, highlighting the need for security testing of forensic tools.	
Bharadwaj & Singh (2018)	HDD	Proposed a region-based random sector sampling method to efficiently search for target data traces in large storage devices, quantifying the necessary percentage of random samples for effective examination.	
Shey et al. (2018)	SSD	Proposed a non-invasive side-channel method using current measurements and machine learning to infer SSD TRIM operations with over 99% accuracy. Analyzed SSD current signatures and recommended a minimum sampling frequency for detecting TRIM operations, validating the current probe technique.	
Jeong & Lee (2019)	HHD SSD USB	Introduced a new forensic signature for tracking storage devices, addressing limitations in HDD and SSD tracking. Introduced unidentified artifacts in UEFI firmware images independent of the operating system and developed a methodology for device tracking based on storage type.	
Kumar (2021)	SSD	Examined challenges in SSD forensics, particularly the impact of TRIM function and background garbage collection on deleted data recovery. Discussed uncertainties in SSD data acquisition and provided guidelines and recommendations based on experimental analysis.	
Chhajed (2024)	HDD SSD	Explored how different operating systems, file system types (e.g., ext4, NTFS, FAT), and storage devices (SSD and HDD) affect the persistence of deleted files. Aimed to improve digital forensics reliability by analyzing the impact of common user activities on the overwrite speed of deleted files..	

In recent years, volatile device data detection has taken a dominant position in the research on trace detection at the storage device layer. This is primarily due to the high investigative value of the data stored in the system’s random access memory (RAM), as well as the rapid development of fileless malware. Fileless malware is more likely to evade traditional detection methods on hard drives and does not leave traces on the disk (Khalid et al., 2023). However, when these malicious programs’ characteristic information is stored in RAM, their hidden data traces can be extracted from memory in an unencrypted and easily accessible format. Therefore, related research is actively exploring the optimal methods for volatile memory detection.

However, these studies also reveal a significant technological imbalance, resulting in insufficient attention to the study of data trace characteristics in persistent storage devices. A possible reason for this is that the data in persistent storage devices is typically based on the structural data of the operating system and file system, making it more appropriate to conduct data trace detection research at the file system and operating system stages rather than at the storage device stage. Additionally, the time sensitivity and complexity of memory, as well as the ongoing debate regarding the credibility of memory snapshots, have long been challenges in memory data investigation. The rapid evolution of fileless malware also makes it difficult for existing memory forensics methods to adapt to new attack techniques. Therefore, although memory forensics holds significant potential in combating modern malware, the technology’s refinement and adaptability still require continuous testing and improvement in practical applications.

Limitations and future directions

Limitations

Although the metrological data trustworthiness detection system holds an optimistic view towards the integration of data traceability detection technology, the analysis in the previous section reveals certain limitations in the existing detection technologies. Metrological trustworthiness detection requires comprehensive verification across the entire “design-production-inspection” chain, involving the coordination and data transmission between metrological devices across different systems. However, current traceability detection technologies often struggle to achieve cross-platform correlation analysis, which limits their application in complex systems. Furthermore, real-time trust verification of dynamic data streams faces significant bottlenecks: existing data detection technologies are inadequate to meet the real-time detection requirements in certain metrological scenarios, where millions of data points are generated per second, resulting in verification gaps for critical data segments. Finally, although the focus of this study is on the local storage process, many data traceability techniques in fields such as network forensics (Mishra et al., 2021) and cloud forensics (Purnaye & Kulkarni, 2022) remain to be further explored.

Future directions

To facilitate the exploration of data trace detection technologies that contribute to trustworthiness verification in the field of metrology, future research can be conducted in the following directions: Smart algorithm-driven trusted verification of measurement data: Measurement data is voluminous and structurally complex, containing various forms of data that form intricate tree-structured data networks. Traditional manual data detection techniques are no longer sufficient to meet the requirements of trusted measurement data verification. The development of automated anomaly detection models by integrating deep learning and artificial intelligence technologies will be an important research direction to enhance the accuracy and efficiency of detection.

Measurement device data tampering forensic technology: As malicious techniques for tampering with measurement device data become increasingly sophisticated, especially with embedded systems in measurement devices, their volatile storage devices hold significant value for data trace detection. Therefore, developing targeted data detection technologies will be a key pathway to addressing tampering issues in measurement devices related to public services, such as fuel dispensers and electric meters.

Multi-modal measurement data full-chain traceability framework: In trusted verification of measurement data, the traceability of data at a single stage is insufficient to establish the trustworthiness of the entire chain. Researchers need to explore collaborative analysis mechanisms for multiple types of information, integrating data content, file system metadata, system logs, and other multi-dimensional data traces, to construct a trustworthiness verification system covering the entire lifecycle of data generation, transmission, and storage.

Conclusion

The integration of data trace analysis into metrology represents a key advancement in addressing the evolving challenges of data trustworthiness in the era of digital transformation. This study highlights the critical role of data traces—residual artifacts from the processes of data generation, transmission, and storage—in detecting tampering and verifying authenticity. By reviewing data trace detection techniques in stages and adapting them to the context of metrology, the study provides a detailed examination of the contributions and limitations of each phase. The research demonstrates that, although techniques such as timestamp consistency checks, deep learning-based anomaly detection, and volatile memory analysis show considerable potential in the metrology field, they are still constrained by the unique complexities of this domain, including the specialized storage architectures of metrological devices and the stringent compliance requirements for data uncertainty propagation as defined by metrological standards. Future research should focus on three key areas: developing automated verification models to handle large-scale heterogeneous data, advancing real-time forensic techniques for volatile memory in embedded devices, and building multi-modal frameworks to integrate trace data across the entire data lifecycle. Ultimately, through the study of data traces, the development of trustworthy detection for metrological data will take a more solid step forward.

Supplemental Information

Supplemental Information 1 PRISMA checklist.

Supplemental Information 2 PRISMA flow diagram.

Additional Information and Declarations

Competing Interests

The authors declare that they have no competing interests.

Author Contributions

Zhanshuo Cao conceived and designed the experiments, performed the experiments, analyzed the data, performed the computation work, prepared figures and/or tables, and approved the final draft.

Boyong Gao analyzed the data, authored or reviewed drafts of the article, and approved the final draft.

Zilong Liu conceived and designed the experiments, performed the experiments, performed the computation work, authored or reviewed drafts of the article, and approved the final draft.

Xingchuang Xiong performed the experiments, authored or reviewed drafts of the article, and approved the final draft.

Bin Wang performed the computation work, prepared figures and/or tables, and approved the final draft.

Chenbo Pei analyzed the data, performed the computation work, prepared figures and/or tables, and approved the final draft.

Data Availability

The following information was supplied regarding data availability:

This study is a review article and does not include any associated data or code.

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
