# Peer review of "Data trace as the scientific foundation for trusted metrological data: a review for future metrology direction"

_PeerJ Computer Science, doi:10.7717/peerj-cs.3106_

## Round 0.1 · original submission · Major Revisions

In this shape, the paper is not publishable. However, the reviewer #2 #3 and #4 gave several points to improve the quality of the paper. One of the common concerns refers to the formatting of the references, and some mistake in reference to the correct figure number, which is the base for getting a paper published in any journal.

I suggest revising the manuscript.

Reviewer 1 ·

Basic reporting

if it is a review paper, then the author should mention Review paper in the title,
The author did not explain the current solution of detection technique in the image.
lack of technical writing and presentation

Experimental design

fail to meet the standard

Validity of the findings

minimum evaluation in the paper.

Reviewer 2 ·

Basic reporting

N/A

Experimental design

N/A

Validity of the findings

N/A

Additional comments

----------- Overall evaluation -----------
The authors present a technical research paper with relevant topic, proper research methodology and potentially good contribution to the field of studies.
The authors are encouraged to resubmit the paper with more clarity on presented performance assessment metrics with the selected relevant Case studies
and possible application scenario with assessment metrics. The paper should be written in proper format, figures should fit within the text, use of font
should be uniform in all paper, as well as references should be updated with most recent results.
Suggestion and Recommendation:
1. Authors may elaborate more on the novelty/contribution of their work and how it
2. Authors need to be specific about their problem statement and the scope of their research.
3. Abstract: elaborate more on the problem statement, findings, and contributions.
4. Introduction is not clear. Authors may contribute more towards this.
Contributes to the literature in the second last paragraph of the introduction clearly.
5. Thorough proofreading is recommended.
6. Afew of the figures are taken from the sources and are not cited properly, either they may be cited properly with permissions or may be removed/
redrawn.

7. The conclusion is not clear and needs revision and clarity and alignment with the abstract and title.
References:
1. Your references are not listed in good style, as citation style is different from one paper to other.
2. some of your references are not complete please check.
3. Some citations (references) created in wrong manner (Please follow journal's criteria).
Authors are encouraged to base on recent references about the current development in blockchain technology. Moreover, technology collaborates with
other technologies to create new paradigms, such as artificial intelligence, such machine learning, deep learning, with federated learning.

Reviewer 3 ·

Basic reporting

The article must conform to professional standards of presenting their work. There are minor grammatical errors that need to be corrected. There is scope for language improvement. The syntax needs to be corrected, e.g., lines 111, 190, 196.

The authors need to add the literature that has been carried out so far specific to metrology. The authors can also add cases where metrology data has been tampered with or affected.

All the figures in the manuscript are not clear; they need to be redrawn and should have a good resolution. The clarity/visibility is very poor. Please draw professional figures. Also, follow a uniform style of figure with uniform coloring, fonts, shapes, sizes, lines, line weights, etc.
Figures in the text do not match the actual figure, e.g., line no. 164; the authors have referred to Figure 2 when it should have been Figure 4 instead.
There is no consistency in naming the "Figures" throughout the length of the manuscript; in some places, figures are referred to as "Fig," while at other places, "Figure" is used. Please check with the journal guidelines to ensure consistency.

The paper needs a lot of work in terms of its presentation and organization. The sections need to be restructured.

The abstract can be improved considering the standard of the journal. The novelity can be highlighted in a better manner. The abstract seems more generic. The authors should clearly include the research objective(s), method(s), conclusion(s), or key findings to enhance the clarity of your work.

There is a lot of redundancy in the Introduction section that needs to be rectified. The paper seems too generic, and I didn't find anything specific to metrology. The authors have collected generic literature about multimedia forgery, tampering, etc. The authors have tried to put in information that is general knowledge. The works with respect to metrology are missing. The authors need to add the works that have been carried out so far specific to metrology. The authors can also add cases where metrology data has been tampered with or has been affected.

Experimental design

The organization of the work is poor and needs to be revised. A lot of work is needed to make the manuscript publishable.

The authors need to perform more investigation to add the technical content specific to metrology. More content regarding the metrology needs to be included; only the term metrology has been used at several places in the text. The content is not sufficient.

Although the authors have used paragraphs and subsections, the manuscript in this structure cannot be published. The paper needs a thorough restructuring. Kindly refer to the review papers that are available in Vogue.

The title of Section "RELATE CONCEPTUAL KNOWLEDGE" should be corrected, e.g., "Related Conceptual Knowledge" or "Related Work" or "Preliminaries."

The "Key Challenges" section needs to be reconsidered; it has been written in a generic manner.

The section "ACKNOWLEDGE" should be "ACKNOWLEDGMENT."

Validity of the findings

The "Conclusion" section needs to be extended considering the length of the manuscript to make it more concrete. The authors
The section "Future Directions" also needs to be revised.

Additional comments

1. Some parts of the text are vague or too generic, e.g., lines 176–177; the authors need to be specific about what "indicators" or "critical facts" they are referring to.

2. The authors should know the difference between "internet" and "Internet," and the term "internet" needs to be replaced by "Internet" wherever applicable in the manuscript, e.g., line no. 195.

3. The tables are captioned at the top rather than at the bottom; it needs to be rectified.

4. The timeline of literature can be given in a chronological order with respect to the year of work in each table.

5. In lines 138 to 142, the authors claim the nature of digital data is volatile; however, not all data is volatile; the data in RAM/caches/registers is volatile, while the data in hard disks/flash disks/SSDs is non-volatile. It indeed is a complex process; however, there is a proper digital forensics method to retrieve/acquire this data, e.g., creating an image of the original data and using write-blockers to prevent any changes (to data or metadata) during the acquisition or analysis process. This is basic knowledge in the field of digital forensics. The authors can identify the problems regarding the tampering that arise (if any) despite the preservation measures taken by forensic investigators or first responders.

6. In the section "Storage device trace detection," the authors have included only the works regarding "physical memory" when the title says "storage device trace detection.".The storage devices implicitly include persistent storage like HDDs, SSDs, and Flash storage. The works related to storage devices are typically missing. Therefore, the authors can divide the section into two parts, viz., persistent storage and volatile storage, and add the work regarding persistent storage.

7. The authors have used the term "reliability" throughout the length of the manuscript without defining "reliability in terms of what parameters." Similarly, the key terms like trustworthiness, data trace, and integrity need to be defined before being used in the paper.

8. A table can be made where the authors can include cumulatively what parameters have been or are tampered with at different stages or what are the potential targets for adversaries.

9. The authors can add a pie chart or a bar graph to include the statistics of papers/literature used in this work to increase the visualization.

Reviewer 4 ·

Basic reporting

This paper addresses an important challenge in the digital transformation of metrology by introducing data traces as a scientific basis for the trusted detection of metrological data. The focus on analyzing the formation and evolution of data traces during different storage stages and the review of tampering detection research is a valuable contribution. However, to strengthen its impact, the paper should clearly define key concepts such as 'trusted detection' and 'data traces' in the context of metrological data. A more detailed explanation of the methodologies for analyzing data traces and detecting tampering would enhance clarity and reproducibility.

Abstract: The abstract effectively introduces the problem, objective, and contributions but lacks methodological details, key findings, and practical implications. Adding these elements would enhance its comprehensiveness and clarity.

Methodology: Need more elaboration for each stage stated. Line 164 - The figure referred to is Figure 4, not Figure 2.

Analysis of existing research: An informative table summarizing key contributions in the field is provided, effectively presenting an overview of relevant studies and their respective contributions. However, please include a critical analysis or key findings ((what should be highlighted from this analysis?) from each referenced work to enhance its value further.

References: Typically, researchers should provide at least 50 recent references for review papers, but some journals require at least 30 references. This paper had 49 references. However, only 13 out of 49 (26%) references are recent (2021-2025). Please add more recent references.

Experimental design

Methodology:
The research methodology provides a well-structured approach to organizing and summarizing research on data traceability; however, it includes more details on the literature search process, such as the number of papers retrieved from each database, the specific filtering criteria applied, and the step-by-step selection process. By doing this, it would enhance transparency and reproducibility. In addition, providing a summary table of this process would offer a clear and concise overview, making it easier for readers to follow and understand the methodology.

Below are the improvements that researchers should consider:
- Provide information on how the literature search is done by stating the critical information such as the keywords used, literature search timeframe and the number of papers per database. This information is important to claim the transparency of your research work.
- The methodology mentions screening studies for relevance and quality but does not clarify who performed them, specific inclusion/exclusion criteria, or quality assessment methods.
- Provide data extraction and analysis details, such as what specific themes were analyzed and methods or tools used for keyword/statistical analysis.
- A brief definition of each stage and an example of categorization criteria are suggested to help readers understand the stage-based approach used in this study.

Validity of the findings

The paper provides a valuable overview of existing research; however, clarifying the criteria for selecting studies and evaluating their quality would be helpful to strengthen its validity. In addition, a more critical comparison of findings, including potential biases or gaps in the literature, would enhance the objectivity and depth of the review.
(This can be achieved by improving the methodology section - refers to comments on the methodology)

Additional comments

Overall, this paper contributes knowledge on data traces and trusted detection for data security and trustworthiness in the field of metrology. However, the contribution is not clearly presented due to a lack of elaboration on the methodology and analysis.

---

## Round 0.2 · Minor Revisions

Dear Authors,

Thank you for addressing the reviewers' concerns in the revised version of your manuscript. The reviewers acknowledge that the paper makes a meaningful contribution by highlighting how digital forensic principles can enhance trust, integrity, and reliability in measurement systems. However, they have indicated that several issues still need to be addressed before publication:

- Revise the abstract to include a clear problem statement, objectives, methodology, key findings, and future directions, to better communicate the scope and contributions of the paper.

- Incorporate relevant technical content on metrology and explicitly explain how trusted metrological data connects to the core arguments of the review, aligning more closely with the manuscript’s title and aim.

- Increase the font size and improve the resolution of Figure 2. The image should be clearly legible at 100% zoom without requiring enlargement.

- Clarify the Y-axis labeling of Figure 6, which is currently ambiguous, and ensure the publication year scale is understandable and accurate.

- Address minor formatting issues, such as missing spaces after punctuation (e.g., line 291).

- Complete and correct all reference entries, particularly the citation for Gary Kessler’s 2007 conference paper.

We look forward to receiving the revised manuscript addressing all these issues.

Reviewer 1 ·

Basic reporting

I did not see a significant contribution in this paper.

Experimental design

The research design is unclear.

Validity of the findings

There is no validation/evaluation of the existing methods

Reviewer 2 ·

Basic reporting

-

Experimental design

-

Validity of the findings

-

Reviewer 3 ·

Basic reporting

1. Clear and unambiguous, professional English used throughout.
Answer: Yes, clear and unambiguous, professional English is used throughout.

2. Literature references, sufficient field background/context provided.
Answer: The authors have provided sufficient literature and context with respect to digital forensics.

3. Professional article structure, figures, and tables. Raw data shared.
Answer: The authors need to improve the quality/resolution of Figure 2. The font size needs to be increased for the text in Fig. 2. The resolution is poor when viewed at 100% zoom. The readers should not need to zoom the figure beyond 100%. Furthermore, regarding Figure 6, I do not understand the publication year scale of the plot. What do the numbers 0, 5, 10, 15, and 20 mean in the Y-axis of this plot? It seems like the number of papers. I believe it needs to be rectified for clarity.
In line 291 of the PDF file, a space after the full stop is required. Please check for similar mistakes.

4. Is the review of broad and cross-disciplinary interest and within the scope of the journal?
Answer: This work falls within the scope of this journal.

5. Has the field been reviewed recently? If so, is there a good reason for this review (different point of view, accessible to a different audience, etc.)?
Answer: While numerous review articles exist within the domain of digital forensics, this particular review extends the field’s relevance by exploring its application in metrology. Integrating digital forensic techniques into metrology represents a meaningful advancement toward enhancing the reliability and accuracy of measurement science. Therefore, this review contributes more significantly to the advancement of metrology than to digital forensics itself.

6. Does the Introduction adequately introduce the subject and make it clear who the audience is/what the motivation is?
Answer: Yes, the Introduction section adequately introduces the subject and makes it clear who the audience is/what the motivation is.

7. Formal results should include clear definitions of all terms and theorems, and detailed proofs.
Answer: Not Applicable

Experimental design

1. Article content is within the Aims and Scope of the journal and article type.
Answer: Article content is within the Aims and Scope of the journal and article type.

2. Rigorous investigation performed to a high technical & ethical standard.
Answer: Since it is a review paper, yes, sufficient investigation has been performed to a high technical & ethical standard.

3. Methods described with sufficient detail & information to replicate.
Answer: Not Applicable

4. Is the Survey Methodology consistent with a comprehensive, unbiased coverage of the subject? If not, what is missing?
Answer: Yes, the Survey Methodology is consistent with a comprehensive, unbiased coverage of the subject.

5. Are sources adequately cited? Quoted or paraphrased as appropriate?
Answer: Yes, the sources are adequately cited and quoted or paraphrased appropriately.

6. Is the review organized logically into coherent paragraphs/subsections?
Answer: Yes, the review is organized logically into coherent paragraphs/subsections.

Validity of the findings

1. Is there a well-developed and supported argument that meets the goals set out in the Introduction? Answer: Yes, there is a well-developed and supported argument that meets the goals set out in the introduction.

2. Does the Conclusion identify unresolved questions/gaps/future directions?
Answer: Yes, the Conclusion identifies unresolved questions/gaps/future directions.

Additional comments

The authors have made commendable revisions to the manuscript, and the paper was an engaging read. By focusing on the domain of metrology, the authors take a valuable step in integrating concepts from digital forensics to enhance this field. The manuscript effectively underscores the critical importance of trustworthiness, integrity, and reliability in metrology- attributes that are essential across a wide range of critical sectors and infrastructures. Given that the fundamental architecture of digital systems (including ROM, RAM, storage, I/O, and network components) is generally consistent across devices, the core principles of digital forensics are broadly applicable, from large-scale servers to compact embedded systems and IoT nodes. This paper offers a well-structured review that can serve as a useful reference for researchers and practitioners in metrology when addressing concerns related to data integrity, authenticity, and potential tampering.

Reviewer 4 ·

Basic reporting

Overall, the authors have made amendments based on the comments provided by the reviewers. With the improved structure, the contributions of the research are well presented. However, some revisions are still needed. The literature review related to metrology is insufficient, especially since it is the main context of the article. Improvements are also needed in the abstract, where the authors should clearly include the research objectives, methods, conclusions, and key findings to enhance the clarity of the work.

Experimental design

The revised version of the manuscript clearly presents how the study was carried out.

Abstract – Ensure that the abstract includes the problem statement, aim/objectives, methodology, key findings/contributions, and future directions.

Literature Review – Insufficient LR on metrology. Include research/technical content related to metrology and explain the relationship between trusted metrological data and the overall content of the study. (Please align with the title of this manuscript.)

Validity of the findings

The findings are clearly presented.

Additional comments

References – Line 491 (paper by Gary Kessler): Please complete the reference. You may try accessing the following link: https://www.scopus.com/record/display.uri?eid=2-s2.0-84867717801&origin=inward&txGid=c986f70d136aea9f78f7c9f8e61349b1

The article is a conference paper published in 2007 in the 5th Australian Digital Forensics Conference, held in Perth, WA, Australia, on 3 December 2007. Conference Code: 93375. Pages: 1–7.

---

## Round 0.3 · accepted · Accept

Thank you for your contribution to PeerJ Computer Science and for systematically addressing all our suggestions. We are satisfied with the revised version of your manuscript and it is now ready to be accepted. Congratulations!